# The Significance of Aggregation Methods in Functional Group Modeling

Huan Zhang [1], Herman H. Shugart [2,*], Bin Wang [3] and Manuel Lerdau [2]

1   School of Landscape and Ecological Engineering, Hebei University of Engineering, Handan 056038, China; huanzhang_bjfu_fe@163.com
2   Department of Environmental Sciences, University of Virginia, Charlottesville, VA 22901, USA; mtl5g@virginia.edu
3   Environmental Sciences Division, Oak Ridge National Laboratory, Oak Ridge, TN 38730, USA; wangb@ornl.gov
*   Correspondence: hhs@virginia.edu; Tel.: +1-434-249-6676

**Abstract:** The growth of forests and the feedbacks between forests and environmental changes are central issues in the planetary carbon cycle, global climate change, and basic plant ecology. A challenge to understanding both growth and feedbacks from local to global scales is that many critical metabolic processes vary among species. An innovation in solving this challenge is the recognition that species can be lumped into "functional groups" based on metabolic similarity, and these functional groups can then be studied in computational models that simulate ecosystem function. Despite the vast resources devoted to functional group studies and the progress made by them, an important logical and biological question has not been formally addressed, "How do the groupings alter the results of modeling studies?" To what extent do modeling results depend on the choices made in aggregating taxa into functional groups. Here, we consider the effects of using different aggregation strategies in simulating the carbon dynamics of a deciduous forest. Understanding the impacts that aggregation strategy has on efforts to simulate regional-to-global-scale forest dynamics offers insights into both ecosystem regulation and model function and addresses this central problem in the study of carbon dynamics.

**Keywords:** individual-based model; gap model; IBM; functional types; UVAFME model; mixed-species model; Tennessee forests; parsing

## 1. Introduction

Here, we apply a well-established individual-based forest model to investigate theoretical issues involving the consequences of grouping data from different species of trees into functional groupings or plant functional types (PFTs). Such groupings are often essential to the initial construction of simulation models of forest ecosystems over time [1,2]. Such functional groupings allow for the development and implementation of ecosystem models with intermediate complexity. The diversity of taxa in most ecosystems is so great that species-level models are far complex to be tractable, however, treating all green matter as functionally identical leads to systematically incorrect conclusions in static and, especially, dynamic models. As discussed in detail below, species richness militates an enormous amount of specific empirical knowledge. This can produce immense model complexity in models of diverse ecosystems. At the other end of the spectrum, the "big leaf" approach, scaling up from the physiology and biophysics of canopies [3,4], misses very important threshold responses that arise from qualitative differences among plants in form and function. Upscaling biophysical and physiological mechanisms is bedeviled by the problem that there is no coherent theoretical basis for treating a multi-species canopy as having system-level optimization properties, as there is for single plants [5].

This problem of aggregation protocol's affecting model results transcends current efforts in global ecological research and has affected ecology before the rise and spread of large-scale models. In an early review of systems analysis in ecology in 1970, Dale [6] noted that, "*One of the most neglected problems in systems studies is the choice of the entities or parts which compose the system. It is commonly assumed that these are self-evident; yet the arguments which have taken place in areas such as the classification of organisms or vegetation concerning sampling, description, and measures of similarity suggest that this is not true.*" This is no less the case today. Ecological systems studies typically begin with a *lexical phase* when the "parts" of the system are defined, followed by a *parsing phase* evaluating the nature of the system relationships [6]. In individual-based models (IBMs) of which the gap models discussed here are a sub-category, there is an explicit recognition that parsing a forest's dynamics as the collective attributes of its individual trees is strongly motivated in model formulation for landscape-to global-scale applications.

The immense diversity that can be found in many forests, especially wet tropical ones, in which there can be over a thousand species in just one square kilometer, has led to the development of aggregation approaches. To address the challenge posed by Dale 50 years ago [6], we conduct a model experiment on the consequences of the lexical determinations of functional types of trees in carbon/biomass models. We do so by using an individual-based model of forest dynamics with parameters derived from the silvics for each species. Using this model as a base, we then use a range of aggregation strategies to test for the effect of aggregation of species into functional groups. Our method is to use an IBM as a source for forest-like information and test the consequences of aggregation of species data into parameterization of functional types. In other words, we are asking how the dynamics of forest biomass as estimated from a long-established and well-validated IBM are altered by aggregating species attributes into functional type attributes and, most importantly, how different aggregation strategies alter these estimations.

The aggregation problem has two elements. The obvious one is that how one chooses to lump species may affect the interactions among aggregates and thus the model output. We refer to this as the inter-aggregate element. The second involves how one pools the trait values for each member of the aggregate. That is, are species trait values to be weighted equally or weighted unequally based on the biomass of species over different time intervals in the dynamic response of the forest simulation. Because the relative biomass of each species changes over the course of a dynamic change during forest succession, the time (relative abundance) at which biomasses of species are determined has the potential to change the specific parameter values for each functional group. We refer to this as the intra-aggregate element. In the experiments described below, we test both elements by employing a frequently used individual-based model in a deciduous forest of low-intermediate diversity. We examine multiple aggregation strategies and the impacts of using different points in succession to develop weightings on model output, and we place these results in the context of the uncertainties associated with remotely sensed (satellite) data that are often used to calibrate/validate ecosystem models.

## 2. Materials and Methods

### 2.1. Model Formulation

The model used—the "University of Virginia Forest Model Enhanced" (UVAFME)—is an object-oriented version of the individual-based gap model FAREAST [7] that simulates forest demography by integrating processes across scales from individual and community to the system. With parameters of species-specific silvics and traits with respect to allometry, growth, mortality, reproduction, and their environmental responses, UVAFME simulates annual growth in terms of diameter at breast height (DBH), regeneration, and mortality of each individual and emergent system-level biomass production while accounting for their interactions with abiotic factors of radiation, temperature, nutrient, and water, as well as disturbances of fire and wind. Specifically, regeneration is simulated by randomly selecting species from a candidate species pool that can survive in a given year

based on the relative size of the seedling bank constrained by environmental factors. Because it is a stochastic model, UVAFME uses a Monte Carlo approach to simulate hundreds of independent patches each of 500 $m^2$. The mean behavior of these patches is assumed to characterize the compositional patterns for a forested landscape corresponding to a shifting-mosaic steady-state landscape under specific environmental conditions [8,9]. Versions of UVAFME have been used to simulate forest ecosystems in the Appalachian Mountains, Rocky Mountains, and the boreal forest ecosystems of Alaska and Russia to determine forest responses to environmental factors such as fire, climate change, insect outbreak, and tropospheric ozone [8–13].

## 2.2. Forest System

UVAFME was applied to a temperate forest system in the Southern Appalachian Mountain region in Eastern Tennessee, USA. 32 species, including 28 deciduous angiosperms and four evergreen coniferous gymnosperms, were dominant in this region [14]. We parameterized the model with these 32 species for the 25 parameters describing species-specific silvics (see Supplement S1 for species parameters in Wang et al. [15]. In the simulation, each of the 200 simulated plots is driven by a stochastic generator for monthly temperature and precipitation. Each simulated plot has the same stochastically generated climate record for each plot each year. The set of simulated plots are analogous to an actual plot inventory system remeasured annually on a landscape that is large enough so that the "plots" do not influence one another, but small enough that the climate conditions across all the "plots" are the same each year. While the simulated weather is the same across the landscape, it varies from year to year as driven by a stochastic weather generator. This stochastic generator is parameterized for the meteorological forcing of temperature and precipitation based on monthly maximum and minimum temperature and mean precipitation and their standard deviations (see Wang et al. [10,15]. As is the case for standard meteorological variables, characterizations of soil chemical features (carbon and nitrogen in the organic and active layers, and soil physical features (soil field capacity, and permanent wilting point are from data sources in the Oak Ridge National Laboratory, Anderson County, TN, USA. These sources are also described in Wang et al. [10,15]. In these simulations, the standard 32-species-based configuration was regarded as a control, against which four different aggregation approaches were examined.

## 2.3. Aggregation and the Creation of Plant Functional Types

We consider a "control" treatment with no aggregation beyond the species level and four "experimental" treatments in which we use different ways to aggregate tree species into plant functional types (PFTs).

Control—None of the species are lumped. All 32 species are included in the simulation. A UVAFME standard model run obtained from the original database species parameters can be used for comparison to other methods to amalgamate species (Figure 1a) into these "experimental" functional groups:

(1) One PFT—All 32 tree species amalgamated into one functional group.
(2) Evergreenness —Lump all 32 tree species into two groups according to the attributes of evergreen or deciduous leaves.
(3) Tolerance—Lump the 32 tree species into five shade-tolerance groups (see Appendix A for details on how tolerance was defined and applied to these species).
(4) Genus—Lump the 32 tree species by genus for the 20 genera represented by the 32 species.

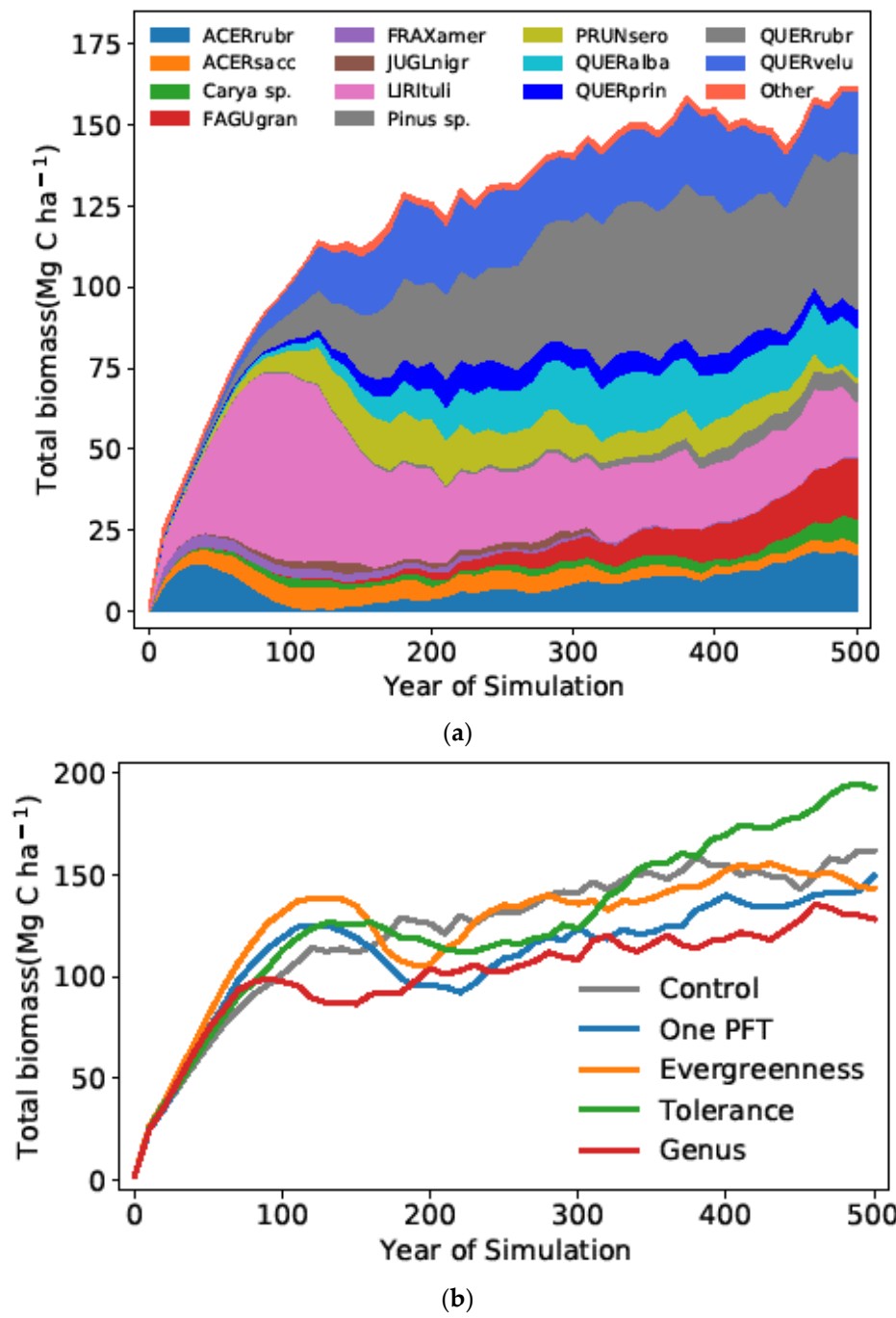

**Figure 1.** (**a**) Dynamics of species-specific biomass (MgCha$^{-1}$) averaged over $200 \times 500$ m$^{-2}$ plots simulated by UVAFME for 500 years. Width of each color band is the biomass of a species or group of species. The topmost line is the total biomass. Species codes are: ACERrubr (*Acer rubrum*); ACERsacc (*A. saccharum*); Carya sp. (*Carya cordiformes, C. glabra, C. ovata, C. tomentosa*); FAGAgran (*Fagus grandifolia*); FRAXamer (*Fraxinus americana*); JUGLnigr (*Juglans nigra*); LIRItuli (*Liriodendron tulipifera*); Pinus sp. (*Pinus echinata, P. strobus, P. virginiana*); PRUNsero (*Prunus serotina*); QUERalba (*Quercus alba*); QUERprin (*Q. prinus*); QUERrubr (*Q. rubra*), QUERvelu (*Q. velutina*); Other (*Aesculus octandra; Cercis canadensis; Cornus florida; Diospyros virginiana; Juniperus virginiana; Liquidambar styraciflua; Nyssa sylvatica; Oxydendron aboreum; Robinia pseudoacacia; Quercus coccinea; Sassafras albidum; Tilia heteophylla*). (**b**) Biomass dynamics (MgCha$^{-1}$) for simulations with four different aggregation functions in an arithmetic mean approach against the standard species-specific simulation.

We used two weighting methods in creating different PFTs. Both were based on aggregating the 25 parameters from all of the 32 species considered to be in each PFT. Such weighting methods could be used in situations in which species-level attributes for species in a given functional type are "lumped" to derive model parameters for a collective "PFT-species" that represents each functional group. For example, the vector of the parameters (representing growth rates, allometry, environmental tolerances, etc.) for all evergreen species could be aggregated and then used in simulations as a "PFT-Pseudo-species" to represent the evergreen PFT. The first procedure to derive PFT parameters is to compute simple arithmetic mean for all 25 parameters across all of the 32 species in a given PFT. Our second method computed the weighted average parameter vector based on the biomass ($MgCha^{-1}$) of the species in each PFT at 25-year intervals of the control simulation. These calculations were made across all species in each PFT for the four different PDF schemes. The essential idea is to compare the effects of constructing PFTs by averaging across species or by weighting the average according to the biomass at a particular time. Four independent simulations were then conducted, each comprising 200 replicate runs starting on bare ground and lasting 500 years.

## 3. Results and Discussion

### 3.1. Inter-Protocol Comparisons: The Effect of Lumping Protocol

The unlumped, species-specific model runs show clear and consistent changes in biomass, with some species traditionally considered "early successional" increasing rapidly and then declining over time, while others consistently increase (Figure 1a). These results are congruent with previous modeling and empirical studies of vegetation change in these systems and have been clearly linked to physiological, allocational, and life-history strategies of each species [16,17]. Similarly, the groups in the four lumping protocols also show a clear increase in biomass averaged across the species in each functional group (Figure 1b). The different simulations typically have local maxima at 100 to 130 years, which is a consequence of the initial even-aged age distributions of the 200 simulated plots transitioning to de-synchronized mixed-aged mosaic across a landscape [18,19]. The increasing biomass levels over the course of the 500-year simulation likely arise from the UVAFME model simulating forest nitrogen dynamics—over the 500 years of stand-development that occurs in these successional dynamics, the soil nitrogen availability increases and with it the potential total biomass. The control case, with each of the species individually considered, is relatively similar to the responses for other functional-types schemes, even though, as one would expect from the many more groups in this case, it is considerably more diverse.

Despite these overall similarities across lumping protocols, there are important differences in the percentage deviations among the four different methods of lumping species relative to the control case (Figure 1b). When the species parameters are lumped to create a single PFT, the biomass level exceeds the control case by as much as 18 $MgCha^{-1}$ (19%) in the simulation; it is below the control by as much as 38 $MgCha^{-1}$ 29% at other times in the 500-year simulation. Comparable ranges of minimum and maximum percentage values at a given year: for the Evergreen PFTs are 30 $MgCha^{-1}$ (32%) maximum relative to the Control and $-22$ $MgCha^{-1}$ (17%) minimum relative to the Control; for the Tolerance-based PFTs 37 $MgCha^{-1}$ and $-20$ $MgCha^{-1}$ (25% and $-14$%, respectively); for the Genus-based PFTs 9 $MgCha^{-1}$ and $-45$ $MgCha^{-1}$ (12 and $-29$%, respectively). These are large values, both in an absolute and in a proportional sense.

To put this variability in the context of measurement uncertainty for large-scale efforts to empirically estimate biomass from remote sensing, the BIOMASS satellite now under construction by the European Space Agency [20] is designed to estimate the biomass for forests to an uncertainty of 20% [21]. That is, the uncertainty in the model output because of the lumping decision is equal to, or larger than, the measurement uncertainty in the satellite data. These large uncertainties are not arguments against lumping choices, but

they do suggest the need to be both (A) explicit about how such choices are made and (B) cautious in interpreting the results from any one lumping choice.

### 3.2. Intra-Protocol Comparisons: The Effect of Weighting within a Functional Group

As a second step, the model parameters for each functional group (as above) were computed as the weighted mean of the species in a functional group. Weighting was based on the mean species biomass on the 200 simulated plots in the control case for a reference year (Figure 2). Hence, species with lower biomass values in a functional group contributed relatively less to the functional group's parameter value than did the species with a greater biomass in a reference year. This procedure mimics using empirical data for species parameters measured in forests of different ages. This approach may be used to circumvent the problems of rare, perhaps ill-adapted, species' skewing the means for a given functional type away from the successful dominant species. This method is also appealing in that it does not involve tabularizing parameters from expert sources but is empirical in its methods of parameter estimation. The weightings are based on species' total aboveground carbon to weight the PFT parameters using the biomass values for each species shown in the control case in Figure 1a. Results are from 200 simulated 500 m$^{-2}$ plots over 500 years with each plot starting from bare soil (Figure 2).

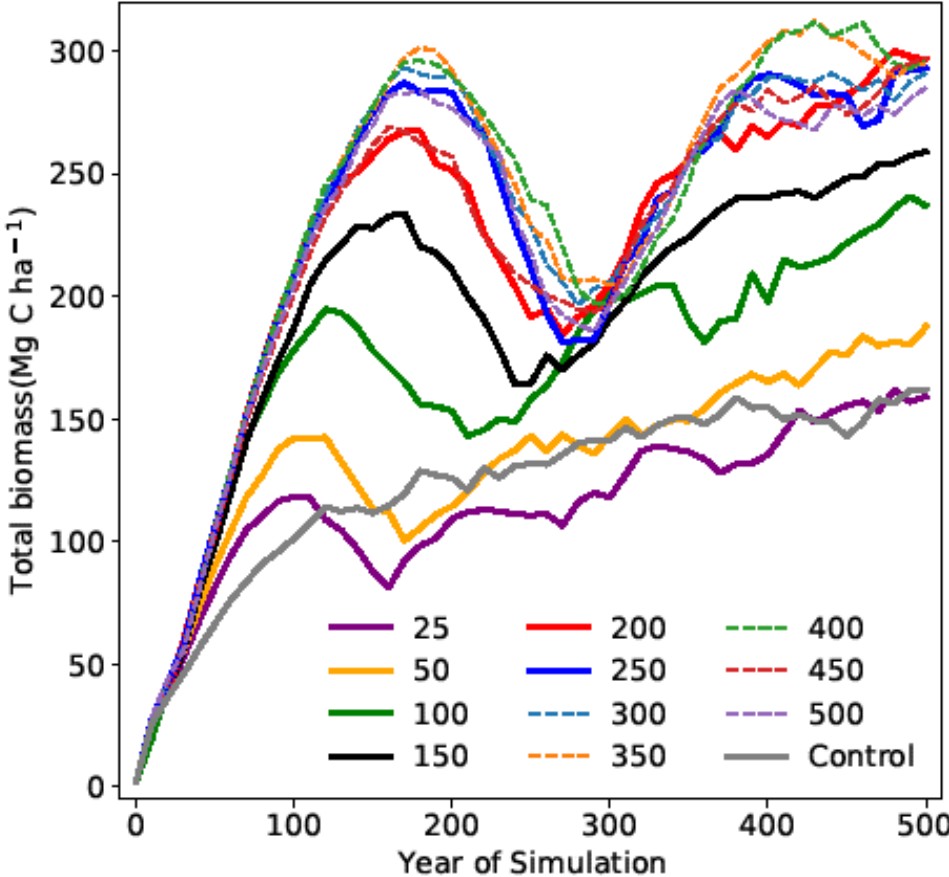

**Figure 2.** Biomass dynamics (MgCha$^{-1}$) with species parameters lumped into evergreeness-based functional types weighted by each individual species' biomass in the indicated reference year. The weightings are based on biomass values of each species shown in the control case in Figure 1a. The control case is also shown as a solid gray line in this figure. Results are from 200 simulated 500 m$^{-2}$ plots over 500 years with each plot starting from bare soil.

Two overarching trends emerge from the analysis of the weighted contributions (Figure 2). First, shown as solid lines, is a successional sequence of forest dynamics from

young (25 years after initiation from bare soil to 250 years after this initiation). The family of curves rises from near zero to a local maximum and it then drops. Subsequently, with additional stand development and soil improvement, the biomass curve rises beyond the earlier local maximum. Models with functional types parameterized from the weighted contributions of years 25, 50, 100, and 150 have relatively reduced biomass values. These results suggest that model outputs from weighted-input approaches depend heavily on the time chosen to determine the weighting factor.

In the second case, for PFT parameterizations based on the weighted biomasses of species in older forests (dotted lines in Figure 2), the simulated forests rise to a maximum biomass value. They then drop as trees in the first cohort suffer quasi-synchronized mortality. Biomass rises again with stand reorganization and the eventual formation of a shifting mosaic forest. Notably, the biomass of forests from PFT-parameterization from older stands reaches values approaching 300 $MgCha^{-1}$, which substantially exceed the maximum value found in the control case (Figure 1a,b). These results emerge from the fact that model runs weighted with data from younger forests tend to overrepresent fast-growing, short-lived species, while model runs from mature forests tend to overrepresent later successional species. In other words, because species proportions change through time, a model parameterized with species weightings from any one time point will only work for that time point. Essentially the age, species composition of the forests that one uses empirically to parameterize a PFT-based IBM can make a considerable difference in model forecasts.

## 4. Discussion

The results from these analyses argue strongly (1) that the methods used to create groups in functional group analysis, or other lumping approaches, have an enormous impact on the model output and (2) the age of the forest at which species contributions are calculated also has a large impact on the model output. Two approaches to these challenges emerge. One involves compiling the critical features of major species, and the second entails developing mechanistic theories regarding tradeoffs in the distinguishing features of the functional groups. The ideal solution would likely be to attempt both simultaneously and iteratively. Taking the complexities in lumping, weighting, and timing into account will improve our ability to model and predict forest dynamics in the face of global change.

As greater amounts of both plot-level and satellite data become globally available, it will become more important than ever to develop and apply robust and explicit protocols for using functional groups in ecosystem models. Recent advances in defining functional group attributes have been an important first step in making such modeling efforts transparent [22]. This paper demonstrates the importance and significance of how such groups are formed and how members are weighted within groups. As more and more case studies emerge from a broader array of systems, it is possible (and to be hoped) that patterns and generalities emerge that will, in turn, lead to further testable hypotheses.

**Author Contributions:** Conceptualization, H.H.S., M.L. and H.Z.; methodology, B.W. and H.H.S.; software, B.W.; validation, H.Z. and B.W.; formal analysis, H.Z., B.W., M.L. and H.H.S.; investigation, H.Z. resources, H.Z.; writing—original draft preparation, H.H.S., M.L. and B.W.; writing—review and editing, H.H.S., H.Z. and M.L.; visualization, B.W.; supervision and project administration, Team effort; funding acquisition, M.L. All authors have read and agreed to the published version of the manuscript.

**Funding:** This work was funded by NSF IOS-2005574 to M. Lerdau, Support from ESUSPI award 80NSSC18K0152 (Development and Validation of Tropical Forest Aboveground Biomass Estimation from BIOMASS Mission Observations), NASA Earth Science Division, USA, and departmental support through the Department of Environmental Sciences, University of Virginia.

**Institutional Review Board Statement:** Not applicable.

**Informed Consent Statement:** Not applicable.

**Data Availability Statement:** Not applicable.

**Conflicts of Interest:** The authors declare no conflict of interest.

## Appendix A. Shade Tolerance Responses

A venerable silvicultural concept is that shade-tolerant species grow and survive better than shade-intolerant species in shaded conditions [23]. Reciprocally, shade-intolerant species have a growth advantage over shade-tolerant species under higher light conditions. This can arise from processes at different scales: ranging from biochemical differences [24]; to leaf physiological differences [25]; to structural differences in canopy geometry [26]. In gap models, different photosynthesis/light curves [27] represent species of different shade tolerance. Shade-intolerant species have higher compensation points and a more rapid rise to *Amax*, the maximum $CO_2$-assimilation rate, with increasing light. The illumination of the canopy profile derives from the canopy leaf-area profile [18].

In this study, shade tolerance of tree species is grouped into five tolerance classes, following the equation developed by [28]:

$$f_L(H) = c_1 * \left\{ 1 - e^{[-c_2 * I(H) - c_3]} \right\} \qquad (A1)$$

where $f_L(H)$ is the light limitation at height, $H$ is the percentage of incoming light, and $c_1$, $c_2$, and $c_3$ are species-specific parameters determining the shape of the productivity-light relationship, among which $c_1$ and $c_2$ involve the asymptote and increase of the productivity–light curve and $c_3$ is the shade-tolerance parameter as described in the ZELIG model [28] corresponding to the light compensation point. $c_1$ ranges from 1.01, 1.04, 1.11, 1.24 to 1.49; $c_2$ from 4.62, 3.44, 2.52, 1.78 to 1.23; $c_3$ from 0.05, 0.06, 0.07, 0.08 to 0.09 [15]. The specific values of these three parameters correspond to species' shade-tolerance classes from 1–5 with 1 representing the greatest shade tolerance. $I(H)$ denotes the available light at height $H$ and is computed according to the Lambert–Beer law (See Supplement S1 [15] for more details).

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
