# Peer review of "The Significance of Aggregation Methods in Functional Group Modeling"

_forests, doi:10.3390/f12111560_

Round 1

Reviewer 1 Report

This study unveils the significance of aggregation methods in functional group modelling. The manuscript is well prepared and timely to highlight the importance of the aggregation methods.

Abstract: Well written.

Introduction: The introduction is very good, the authors demonstrate a thorough knowledge of the published literature, the underlying problem and highlight the importance and background to carry out this investigation.

Material and Methods: Methods are technically strong and well explained

Discussion. It is good but very short indeed.

Author Response

The reviewers comments were positive and brief.  We had tried to take this endorsement and advice in mind in our revisions.

Reviewer 2 Report

The paper presents an analysis of effects of different ways of parameterization of gap forest models. This is an interesting simulation experiment and the analysis of the model itself, but there are some issues about this model type, which would be good to mention in the Introduction and/or Discussion. Gap models appeared already in 80-90ths and are interesting first of all to analyze inter-species and inter-generation dynamics. For this purpose, the aggregation of species into few broad groups can considerably decrease its analytical capacity. For predicting the real forest biomass, the 500-years-long simulation of undisturbed forest growth from bare soil under stable climatic conditions, taking into account that climate is changing and most of forests are managed, the applicability of the model is limited to rather infrequent cases as natural reserves, hardly accessible virgin forests, early successional stages (e.g., abandoned agricultural land) etc. It does not mean that such analysis is useless, but this issues should be addressed. Concerning analyzed aggregation methods, only tolerance has a mechanistic meaning. I would suggest (maybe for future) to use 3-parameters grouping (light/water/nutrients tolerance) or simply to group species by formal cluster-analysis using whole parameter set

The Methods part is too brief. In particular:

  • It would be good to give more information about the model. E.g., are meteorological variables assumed to be constant or varying? How regeneration is partitioned among species? How initial nutrients content in the bare soil is set? Maybe it would be good to add few sentences about gap models in general.
  • The key issue to be clarified in the Methods is how individual species are treated within PFT groups. E.g., if you have spruce and pine joined in the PFT of coniferous, it means that they will have same set of parameters. So, each individual simulated tree will be treated as spruce or pine, or as conifer in general? If the first then their partitioning will be fully stochastic, isn’t it?
  • The weighting of PFT parameters should be clarified. If all species in the given PFT will have the same set of parameters, how can you follow in time the partitioning of biomass between species within PFT (as it mentioned in lines 118-122)? Either you follow in the model individual species with their individual parameter sets and then you can get their partitioning for each year – but in this case where is the place for aggregation into PFT? Or you use the same parameter set for the whole PFT – and then the dynamics of partitioning between individual species within PFT will be fully stochastic. The issue was clarified only in the results (lines 179-182), but I think it belongs to Methods
  • It would be good to formulate more clearly in the Methods, the effect of aggregation on which variables is evaluated (it seems from the Results that it is only total biomass).

Author Response

We have marked the responses to the reviewers comments directly on to the reviewers PDF text and also made these responses in the revised manuscript.  We appreciate the comments and the close inspection of the paper.  
